# Automatic Detection of Microaneurysms in Fundus Images Using an Ensemble-Based Segmentation Method

**DOI:** 10.3390/s23073431

**Published:** 2023-03-24

**Authors:** Vidas Raudonis, Arturas Kairys, Rasa Verkauskiene, Jelizaveta Sokolovska, Goran Petrovski, Vilma Jurate Balciuniene, Vallo Volke

**Affiliations:** 1Automation Department, Kaunas University of Technology, 51368 Kaunas, Lithuania; 2Institute of Endocrinology, Lithuanian University of Health Sciences, 50140 Kaunas, Lithuania; 3Faculty of Medicine, University of Latvia, 1004 Riga, Latvia; 4Center of Eye Research and Innovative Diagnostics, Department of Ophthalmology, Oslo University Hospital and Institute of Clinical Medicine, Faculty of Medicine, University of Oslo, 0372 Oslo, Norway; 5Department of Ophthalmology, University of Split School of Medicine and University Hospital Centre, 21000 Split, Croatia; 6Lithuanian University of Health Sciences, 44307 Kaunas, Lithuania; 7Faculty of Medicine, Tartu University, 50411 Tartu, Estonia

**Keywords:** diabetic retinopathy (DR), image segmentation, microaneurysms (MAs), encoder-decoder deep neural network

## Abstract

In this study, a novel method for automatic microaneurysm detection in color fundus images is presented. The proposed method is based on three main steps: (1) image breakdown to smaller image patches, (2) inference to segmentation models, and (3) reconstruction of the predicted segmentation map from output patches. The proposed segmentation method is based on an ensemble of three individual deep networks, such as U-Net, ResNet34-UNet and UNet++. The performance evaluation is based on the calculation of the Dice score and IoU values. The ensemble-based model achieved higher Dice score (0.95) and IoU (0.91) values compared to other network architectures. The proposed ensemble-based model demonstrates the high practical application potential for detection of early-stage diabetic retinopathy in color fundus images.

## 1. Introduction

According to the World Health Organization (WHO), 422 million people were diagnosed with diabetes worldwide, and this number has steadily increased over the past few decades. In 2021, 537 million adults (20–79 years) had diabetes, which is 1 in 10 individuals. This number is predicted to rise to 643 million by 2030 and to 783 million by 2045 [1]. Diabetic retinopathy (DR) is a diabetes complication that leads to vision impairment [2]. The main cause of impairment are the damaged blood vessels of the light-sensitive tissue called the retina, which is found at the back of the eye. Untreated DR can lead to blindness. Over time, high levels of blood glucose can lead to capillary occlusion and the impairment of blood supply to the eye tissues. The eye attempts to grow new blood vessels due to self-regulatory processes and to compensate this need. However, these new blood vessels can easily break and leak. There are two main stages of DR: early DR (non-proliferative DR; NPDR) and late DR (proliferative DR; PDR). With NPDR, blood vessels in the retina can close off. This blockade is called ischemia. New abnormal blood vessels do not grow in the NPDR type, but untreated NPDR can still lead to permanent vision loss. PDR is the more advanced stage of diabetic eye disease, characterized by neovascularization. New abnormal blood vessels often bleed into the vitreous and form scar tissue that can lead to a detached retina.

The first signs of NPDR are capillary microaneurysms (MAs), dot and blot retinal hemorrhages, as well as hard and soft exudates which look like cotton-wool spots. Capillary MAs appear as tiny red dots scattered in the retina posteriorly. They may be surrounded by a ring of yellow lipid or hard exudates. Hard exudates are the result of vascular leakage. Soft exudates are areas of microinfarction of the retinal nerve fiber layer that lead to retinal opacification. MAs are early symptoms of DR. Moreover, the number or count of MAs is associated with DR severity [3]. Therefore, timely and accurate detection of MAs is important for early treatment and damage reduction caused by DR.

Examination of a patient’s fundus is undertaken by ophthalmologists. Inspection is usually performed on color fundus images by a trained grader, retina specialist or ophthalmologist. It is time-consuming work and results are prone to error. The WHO recommends annual eye screening for diabetes patients. However, timely and accurate DR diagnosis and analysis without automatic screening tools is difficult to execute because of the economic burden on health service providers and limited access to trained specialists in rural or remote places. Computer-aided automatic screening of eye fundus images could be a solution that helps ophthalmologists and health service providers to screen larger population of patients. The yearly growth in the number of DR patients and the limited access to experienced ophthalmologists has motivated the authors of this study to propose a deep learning (DL)-based automatic approach and technique for the detection of early stage of DR, such as MAs.

MAs appear as small, round-shaped, reddish blobs near capillaries in the fundus image. Despite MAs being a symptom of the earliest DR stage, they can be found at all stages of the disease progression. An example of the presence of several MAs is shown in Figure 1. The diameter of MAs is usually between 15 and 60 µm, or somewhat larger. MAs are difficult to detect due to their tiny structure and color similarity to the surrounding tissue. MAs appear in the fundus as red and dark red isolated small blobs with distinguishable contours.

In this study, three known neural structures and a proposed ensemble-based method were trained and tested on a proprietary eye fundus image dataset for the semantic segmentation of MAs.

## 2. Background

Computer-aided automatic screening of color fundus images has been widely applied and has gained an important role in the diagnosis and decision support of different eye diseases, especially DR [4,5,6]. The usage of DL methods and algorithms has been rapidly growing, demonstrating promising results which can be applied in daily diagnostic activities. DL-based models and algorithms are applied to analyze retinal fundus images as part of the main aim to develop automatic computer-aided decision support systems that aid in the diagnosis of DR and other eye diseases [7]. Kevin Raj et al., presented a computer vision algorithm that is used to segment veins and arteries in fundus images. A novel AV-Net neural network was thereby presented, which combines low and high-level features extracted from images using the ResNet-50 backbone. The advantage of their presented method is the lack of need for a segmented vasculature map as training input [8]. Siva Raja et al., applied computer vision techniques for the detection of glaucoma. The technique combined the set of image noise-reducing filters and the ensemble of clustering algorithms [9]. Dharmawan et al., developed a computer-aided system that was used to investigate micro-vessels, vessels with a central reflex and vessels in the presence of lesions. The algorithm used a new blood vessel enhancement method and combined it with an autoencoder U-Net [10]. Different loss functions were investigated to highlight the most useful loss function that could be used to segment the blood vessels. The results presented in their research work [11] have shown that Focal + Dice loss functions provide better results than other functions. Other research groups have proposed fundus image enhancing techniques which are based on a mathematical model for processing fundus images. Experimental investigation has shown that the regional illumination and the refinement strategy can make the diagnostic process more precise [12], [13]. Yen et al., presented a segmentation method for blood vessels: optic disc and cup segmentation. The segmentation solution was based on a multiscale network and dedicated image filters [14]. Yijie et al., were able to improve the segmentation performance of the U-Net autoencoder by using Azzopardi’s and Roxchoudhury’s methods [15]. Other researchers proposed a neural model that can cope with three different perspectives. Their model was successfully applied for lesion detection in the fundus images [16]. Hervella et al., proposed a technique for self-supervised pre-training of the segmentation model. The proposed technique was successfully applied to segment the optic disc and cup in the eye fundus [17]. Tang et al., modified DeepLabv3+ to create a more accurate vessel segmentation method [18], and Wu et al., used pretrained an Alexnet model for the same purpose [19].

The detection of MAs using computer vision algorithms is a challenging task because of many factors, such as visual characteristics of MAs with regard to size, shape, and color, the presence of noise (for example dust on the optics), uneven illumination and low contrast of fundus images. Moreover, there is a limited number of public datasets with accurately labeled DR examples suited for training and testing. Santos et al., proposed an approach that can overcome these problems. The proposed approach was based on image processing techniques, data augmentation, transfer learning and DL networks, such as YOLOv5 [20]. Nawaz et al., applied Inception-v4 model recognition and grading of DR. The authors proposed two configuration modes (fine tune mode and fixed feature extraction mode) for transfer learning in the Inception-V4 model. Both configuration modes achieved a decent classification accuracy [21]. Other researchers used a combination of unsupervised and supervised techniques to solve the MA localization problem. Radon transform and multi-overlapping windows were applied on fundus images for the extraction of important landmarks, such as the optic nerve head and retinal vessels. The localization and classification of the MAs were, thereby, performed using support vector machines [22]. Valizadeh et al., developed a custom convolutional neural network for the detection of DR lesions. Their proposed method was tested with eye fundus images acquired from 80 patients, and the results have shown that the proposed method has a potential to be utilized for automatic screening [23].

However, many computer vision researchers are stating that the detection of microlesions, such as Mas, remains a significant challenging task, and the mentioned methods based on DL have several limitations. First, a significant quantity of data is needed to train and test deep neural networks. In many cases, the required amount of data is not available. The analyses of presented methods only use 2D color fundus images, from which 3D structural information of the eye fundus cannot be extracted. Therefore, discriminating 3D features that can be applied to classify the DR grade are not used.

## 3. Methods and Materials

### 3.1. Dataset

Custom datasets consisting of color fundus images were used in this study. The datasets were created during the execution of international EEA and Norway program project PerDiRe, involving health institutions from Lithuania, Latvia, Estonia and Norway. Several different image-capturing medical devices were used in the study. In general, the medical device Zeiss Visucam 500 was used to capture color fundus images with a resolution of 2124 × 2056 pixels. The dataset consisted of 300 color fundus images (Figure 2 left), from manually segmented images (ground truth annotations), where black blobs indicate the appearance of MAs (Figure 2 right). Ground through images (annotated images) were prepared by a group of ophthalmologists originating from 4 different countries. Each fundus image was carefully inspected and annotated by specialists with more than 5 years of experience. Ground truth segmentation maps had the same size as the original color fundus images. The diameter of the circular area of an MA does not usually exceed more than 50 pixels. Such an area is more than 4000 times smaller than the entire image.

Eye fundus image color composition is affected by age; therefore, in this study, semantic segmentation was performed using deep neural networks. Networks such as U-Net, U-Net++, residual U-Net and an ensemble of all three have limited computational capability, mainly because of computation resources, to use whole original fundus images used as training inputs. The rational image size considering computational power and segmentation accuracy was 576 × 576 pixels. However, direct size reduction of the original fundus image to 576 × 576 pixels means that the MA areas were also reduced 36 times. Such a small area was difficult to segment and there was a high risk that many of the MAs could merge with the surrounding background. Instead of resizing the image, we proposed the use of a cropped region of interest (RoI) of the fundus image, where each RoI was selected using an overlapping sliding window of fixed size. An RoI selection functional diagram is shown in Figure 3.

In our analysis, 30 percent of overlapping was used. The usage of RoIs instead of original fundus images had several advantages. First, the amount of image samples which is needed for training and testing is increased more than 30 times, i.e., up to 10,000 images (RoIs with only black pixels are discarded). Second, the demand of computation resources is lowered. Third, the original size and form of the MA’s blob is preserved. However, the reconstruction of the whole segmentation map from smaller RoIs involves several image processing steps that encounters the overlapping of RoIs (further elaborated in Section 3.3).

### 3.2. Proposed Segmentation Model

In this study, the impact of segmentation accuracy of different structures of deep neural networks is investigated. The U-Net architecture is widely known for its application in the medical domain. For this reason, variants of this architecture were selected: U-Net, U-Net++, Residual U-Net and an ensemble of these networks were chosen for investigation. Since MAs vary in size, this was a key motivation for using encoder–decoder CNN architectures of various depths and complexity. In the following section, the investigated network structures and proposed model are described.

The structure of the U-Net model is shown in Figure 4. The U-Net network is commonly used for image processing tasks where the output and input must have a similar size, and the output needs that amount of spatial resolution [24]. The U-Net-type model is used in this study to generate the MA segmentation mask of the color fundus image. The U-Net-type model comprised encoding (down-sampling), decoding (up-sampling) paths and skip connections. The down-sampling path (left side of the model) consists of a series of two convolutional layers of reduced grid size and max-pooling layer. The down-sampling path extracts the contextual information, while the up-sampling path recovers the spatial information. To output an MA segmentation mask of the same size as the input image in the U-Nets, an up-sampling path is used.

The high-resolution image processed through the down-sampling path is essentially converted into a low-resolution image. In our study, images with a size of 576 × 576 pixels were tested accordingly. The size of the input image was gradually reduced, while the depth was gradually increased, until the image size reached 18 × 18 × 512. Finally, a so-called “bottleneck” was reached in the 5th convolutional layer, which was in the middle between the encoder and the decoder. Convolutional and batch-normalization layers were used in the “bottleneck.” The low-resolution image was decoded to a high-resolution image in the up-sampling path. The skip connections were used to transfer local features by concatenating feature maps from the down-sampling side to the up-sampling side. After each skip connection, two consecutive regular convolutional layers were used. The last output layer was a convolution layer with a filter size of 1 × 1 and a softmax activation function. The output result of the U-Net model was a MAs segmentation mask of size 576 × 576 pixels. This model was compiled with Adam optimization algorithm, with the loss function defined as binary cross entropy (BCE). In total, the U-Net model consisted of more than 7.7 million of trainable parameters.

The architecture of the Residual U-Net model is shown in Figure 5. ResNet34 pre-trained on the ImageNet was used as the encoder to capture deep semantic features (down-sampling path). The decoder consisted of an augmented attention module and a decoder block (up-sampling path).

ResNet is a convolutional neural network (CNN) architecture that consist of residual blocks (ResBlocks) with skip connections, and these blocks are connected in series. Each residual block has two connections from the input. The first input goes through a series of convolutional layers, batch normalization and linear functions. The second input skips over that series of convolutions and functions. The tensor outputs of both connections were added up together, as shown in Figure 6. The experiments have shown that the usage of more layers in a neural network can make it more robust for image variation, at the cost of reduced accuracy. To maintain the accuracy and robustness (or to solve the so-called vanishing gradient problem), residual networks become useful when the complexity of the neural network structure is increasing.

A ResBlock has a 3 × 3 convolutional layer that is connected to the batch normalization layer and an activation function ReLU (rectified linear unit). The ReLU output is again connected to a second 3 × 3 convolutional layer and batch normalization layer. The skip connection (or identity connection) skips the mentioned layers and is added directly before the second ReLU function. This type of residual block was repeated to form a down-sampling residual network. The up-sampling path of the residual U-Net model was similar to U-Net described earlier. The skip connections were used to transfer local information by concatenating feature maps from the residual down-sampling side to the up-sampling side of the model. The original pixel values were concatenated with the final ResBlock through a skip connection to allow final segmentation to take place with information about original pixel values inputted in the model. The output result of the U-Net model was an MA segmentation mask of size 576 × 576 pixels. This model was compiled with Adam optimizer, with the loss function BCE. In total, the ResNet34-UNet model consist of more than 21.6 million of trainable parameters.

The third deep neural network which was used in the ensemble of models was a U-Net++-type network. This network differs from the original U-Net model in three ways: (a) skip connection uses convolutional layers to transfer semantic information from the encoder to the decoder; (b) dense skip connections are used to improve training or the so-called gradient flow and (c) there is network pruning capability through deep supervision [25]. The U-Net++ model was based on both nested and dense skip connections (Figure 7). This model is effective in capturing fine-grained details of 2D images. The black part of the network shown in Figure 7 indicates the original U-Net network, while blue and gray represent convolutional blocks on the skip connections. The usage of more convolutional blocks in the network results in a higher number of parameters. Therefore, more time is needed to train U-Net++-type networks. In U-Net++, deep supervision—Λ is used that allows the complexity of the network to be pruned and adjusted, keeping in mind inference time and performance (segmentation accuracy).

The presented segmentation models have their limits of accurate segmentation when applied on different eye fundus images. These limitations can appear when the eye fundus is not evenly illuminated, has a different image noise that can occlude important image features or the fundus image is acquired using a different image sensor. Each segmentation model produces different segmentation results, which are almost accurate. All models trained on the same dataset are joined into one ensemble to obtain a higher accuracy and more stable results. The functional diagram of the proposed ensemble is shown in Figure 8. The ensemble combines the three previously introduced models, i.e., UNet, UNet++ and ResNet34-UNet models.

### 3.3. Reconstruction of the Predicted Segmentation Map from RoIs

The reconstruction flow of whole segmentation map from individual RoIs is presented in Figure 9. The reconstruction of prediction map consists of three main steps. All individual RoIs are segmented and predictions are acquired using one of proposed segmentation model in the first step. Each predicted RoI is placed on the canvas in the same way they were cropped before and average prediction value from RoIs is calculated. The predictions of the overlapping area are added together and divided by the overlapping number in the second step. Finally, the predicted MAs segmentation map is calculated using a threshold value. The pixels for which predictions were higher than 0.5 were marked with black color; in all other cases, the pixels were marked as white pixels.

### 3.4. Semantic Segmentation Performance Measures

The effectiveness of the semantic segmentation model is measured using intersection over union (IoU) criteria (or the Jaccard index) and the Dice coefficient (F1 Score). Intersection over union is a metric that measures the degree of overlap between two regions of interest. IoU evaluates the overlap of the ground truth (given by the human expert) and predicted regions (produced by the trained model). IoU is computed as the ratio of the overlap area to the combined area of the prediction and ground truth. Intersection over union values are in the range from 0 to 1, where 0 means no overlap and 1 means perfect overlap. This measure is the primary metric to evaluate model accuracy, where comparative analysis is performed pixel-by-pixel. IoU is calculated using Formula 1.
(1)$$IoU=\frac{TP}{\left(TP+FP+FN\right)},$$
where TP is true positive. It is an area of intersection between the segmentation mask (S) (the model output) and ground truth (GT) that is manually made by the expert. TP is calculated as a logical AND operator applied on the areas, as shown in Formula 2.
(2)TP=GT∧S,

FP is false positive. The predicted area is outside the ground truth regions. FP is calculated as logical OR of GT and the difference of the segmentation mask (see Formula 3).
(3)FP=(GT∨S)−GT,

FN is false negative, i.e., the number of pixels in the ground truth region that the segmentation model not predicted. It is calculated using Formula 4.
(4)FN=(GT∨S)−S,

TP, FP and FN terms are used to evaluate segmentation results, because these values correspond to image areas or the number of pixels.

The Dice coefficient (DC) is used to evaluate the performance of image segmentation methods as well as IoU; it ranges between 0 and 1, where 0 means no overlap and 1 means perfect overlap. DC evaluates how similar the ground truth and predicted areas are. DC is calculated using Formula 5. As it can be seen by Formulas 1 and 4, IoU tends to penalize differences between prediction and ground truth, while the Dice score positively weighs matches.
(5)$$DC=\frac{2*TP}{2*TP+FP+FN},$$

## 4. Results

The segmentation performance compared three different neural networks and the ensemble of models. The results of the quantitative comparisons are shown in Table 1, and the results of the segmentation visualization of different methods are shown in Figure 10, Figure 11, Figure 12 and Figure 13 in this section.

### 4.1. Experimental Setup

The experimental investigation of the proposed methods was executed on the images acquired using a Zeiss Visucam 500 medical device. This high-quality system provides relatively high-resolution eye fundus images. In total, 300 annotated images were preprocessed and approximately 10,000 image patches were created (more details about the image preprocessing are given in Section 3.1). The entire dataset consisting of image patches was randomly divided in the following order: 70% was given for model training, 10% for validation and the remaining 20% of data were used for testing purposes. All computation, training and testing were performed using a desktop computer with an Intel i7 CPU and a Nvidia RTX3080 graphical card.

### 4.2. Experimental Results

Figure 10 presents the visualization results of the MA segmentation using three different network architectures and ensemble-based approach. Several random patches from testing dataset are shown. Additionally, from left to right, ground truth annotation and the model outputs are given, accordingly.

The given examples demonstrate the difficulty of the MA detection task. It takes several years for a human expert to master image grading procedures and accurately detect tiny blobs that relate to MAs. Our experimental investigation shows that different segmentation models give different prediction outputs. In some cases, MAs are detected accurately, while in others, these models fail to detect fuzzy edges of the MAs. The proposed ensemble-based approach uses collective predictions from three different models and this approach outperforms individual models (deep neural networks).

The relationship graph between IoU performance values and the tested image is shown in Figure 11, where IoU values are given on the vertical axis, and image ID is given on the horizontal axis. One hundred randomly selected testing images were used to create this graph. The red curve represents the performance of the ResNet-UNet network, the blue curve represents the performance of the U-Net++ network, the green curve is acquired using the U-Net network and the black curve represents the results acquired using the ensemble-based model. On average, the ensemble-based model has an IoU equal to 0.91. This result is the best when compared to other networks used (Table 1). The IoU value of the ensemble-based model drops drastically when all three models fail to accurately detect an MA.

The relationship graph between Dice scores and the tested image is shown in Figure 12, where Dice scores are given on the vertical axis and image ID is given on the horizontal axis. One hundred randomly selected testing images were used to create this graph. The red curve represents the performance of the ResNet-UNet network, the blue curve represents the performance of the U-Net++ network, the green curve is acquired using the U-Net network and the black curve represents the results acquired using the ensemble-based model. On average, the ensemble-based model has a Dice score equal to 0.95. This result is again best when compared to other networks (Table 1). The Dice score of the ensemble-based model drops drastically when all three models fail to accurately detect an MA.

The best segmentation performance of the investigated methods is highlighted in bold (Table 1). The proposed ensemble-based model has the best IoU = 0.91 and Dice score = 0.95 values compared to other models. The lowest performance was achieved using the U-Net++ deep neural model on the proposed dataset; accordingly IoU = 0.76 and Dice = 0.82.

Examples of MA prediction maps of whole fundus image acquired using the ensemble-based model are shown in Figure 13. The ground truth annotations of the real MA are marked in black. The predicted MAs are shown in red (red and black blobs overlap). These examples demonstrate that the usage of the ensemble-based model has immense practical application potential in the early-stage detection of DR.

## 5. Discussion

We hereby propose a method based on the ensemble of models for MA detection in a color fundus image. This method and three other architectures were evaluated on a dedicated dataset. Despite the performance in MAs segmentation, some misclassification can appear. There are several causes for a misclassification. MAs detection and segmentation is a difficult task as there is a disagreement between different experts in the diagnosis of Mas in itself. Therefore, the accurate segmentation is directly impacted by the quality of the dataset provided for training. It should be noted that the same fundus image can be analyzed differently by different human experts (or graders). Some image patches may be labeled as MAs, when there are no clear signs of MAs, and some not, when they have the fuzzy signs of MAs. The erroneous segmentation of MAs can be caused by image noise or artifacts that is recorded in fundus image. In addition, the proposed ensemble-based model achieved better performance, but with higher computational time than three other networks (U-Net, ResNet34-UNet and U-Net++). The training and testing of three independent segmentation models were performed on a GPU graphical card. We used Nvidia GTX1080 Ti with 11 GB of internal memory. All three models were trained on same training dataset with same hyper parameters. It took around 28 h to train the smallest U-Net segmentation model that have 7.7 million of trainable parameters; 57 h were needed to train ResNet34-Unet segmentation model that consist of 21.6 million of trainable parameters. The current model is the largest in ensemble-based approach. It took around 37 h to train U-Net++ segmentation model that consist of 9.1 million of parameters. In total, it took 122 h (5 days) to train all ensemble-based segmentation model. From the viewpoint of a practical application, the proposed MAs segmentation method can be used as an artificial assistant in clinical practice. If MA is detected in the fundus image, the ophthalmologist can focus to the suspicious region of an image and conclude the final diagnosis. Such practical application deserves further development towards computer-aided MAs detection.

## 6. Conclusions

The automatic detection and segmentation of MAs remains a challenging task for computer vision researchers. In this study, we proposed the use of an ensemble-based model to automatically segment MAs by analyzing collective predictions. In addition, we proposed a methodology for image processing in order to obtain a more statistically important dataset and reduce the requirements for computational resources. We conducted an evaluation of segmentation models using a dataset acquired during real clinical screening of the eye fundus. The proposed ensemble-based model demonstrated practical application potential for the detection of early stages of DR. The ensemble-based model achieved higher Dice score (0.95) and IoU (0.91) values compared to other architectures, such as U-Net, ResNet34-UNet and U-Net++. Future research work will involve the development of the segmentation method of other DR signs, such as hemorrhages, soft and hard exudates, the formation of new blood vessels and abnormal vessels. Additionally, extending the current dataset with new fundus images and developing image enhancing techniques is planned, which might better highlight the visibility of DR symptoms. The ethnical bias was not tested in the current research and should be addressed in future work.

## Figures and Tables

**Figure 1 sensors-23-03431-f001:**
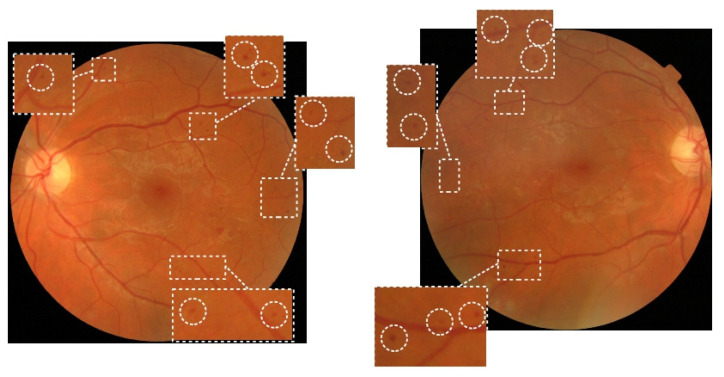
Examples of fundus images of the left and right eye with several microaneurysms.

**Figure 2 sensors-23-03431-f002:**
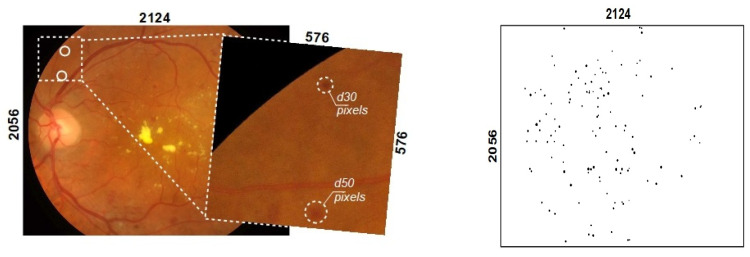
An example of a color fundus image (**left**) and annotated image represented as an MA segmentation map (**right**).

**Figure 3 sensors-23-03431-f003:**
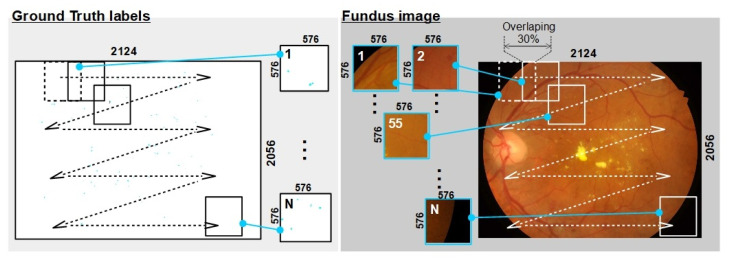
Functional selection diagram of the region of interest based on an overlapping sliding window approach.

**Figure 4 sensors-23-03431-f004:**
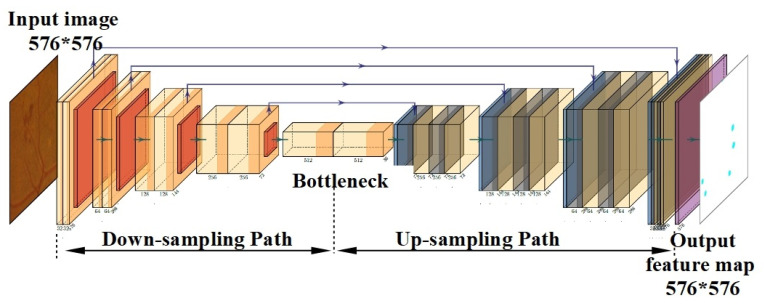
Deep neural network structure of the U-Net model.

**Figure 5 sensors-23-03431-f005:**
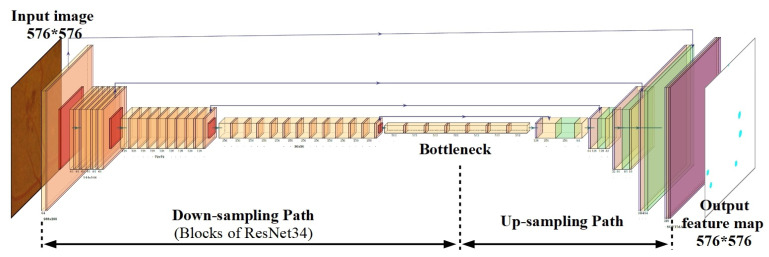
The deep neural network structure of the ResNet34-UNet model.

**Figure 6 sensors-23-03431-f006:**
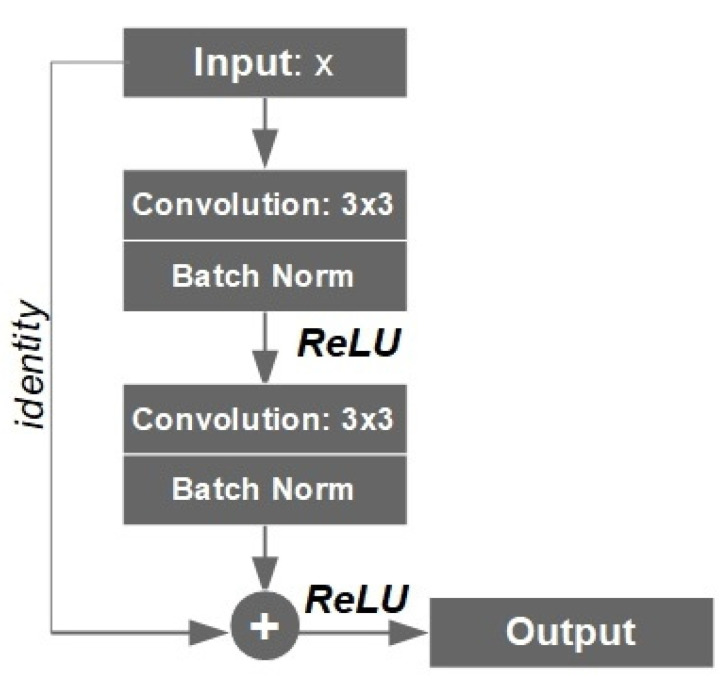
An example of ResBlock within a residual network.

**Figure 7 sensors-23-03431-f007:**
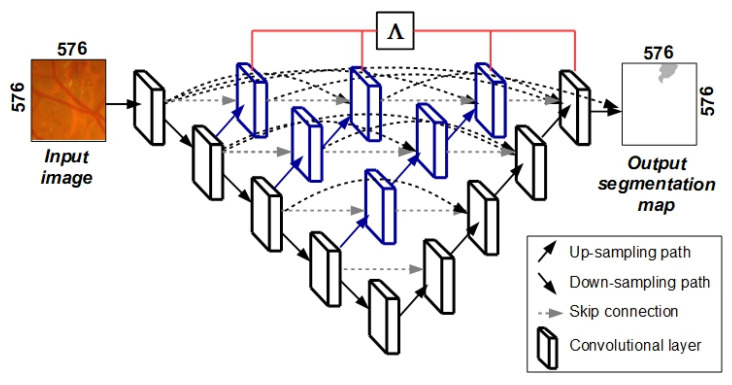
The architecture of the U-Net++ segmentation model.

**Figure 8 sensors-23-03431-f008:**
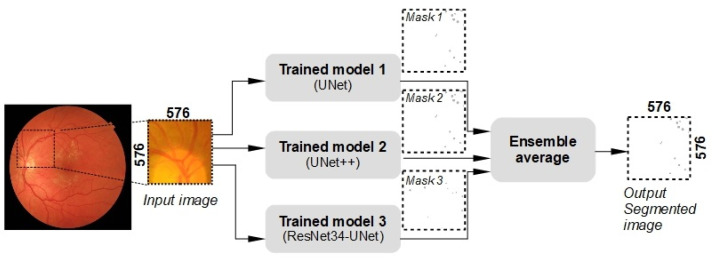
Functional diagram of the proposed ensemble-based segmentation model.

**Figure 9 sensors-23-03431-f009:**
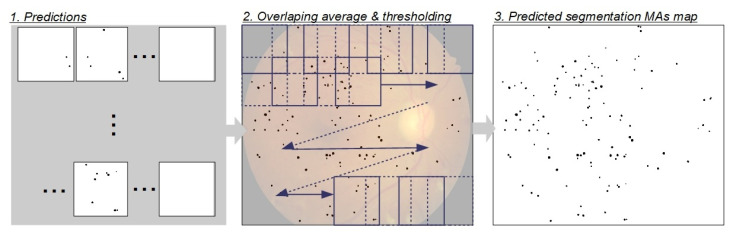
Diagram of the reconstruction of the whole MA segmentation map.

**Figure 10 sensors-23-03431-f010:**
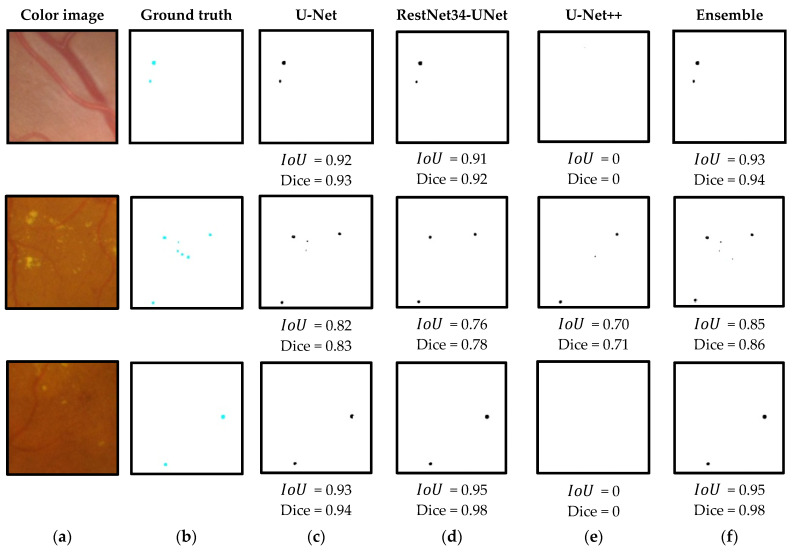
Visualization of the microaneurysm (MA) segmentation results of RoIs acquired from color fundus images: (**a**) original RoI, (**b**) the corresponding ground truth annotation, (**c**) MA segmentation results using the U-Net model, (**d**) MA segmentation results using the residual U-Net model, (**e**) MA segmentation results using the U-Net++ model and (**f**) MA segmentation results using the ensemble of models.

**Figure 11 sensors-23-03431-f011:**
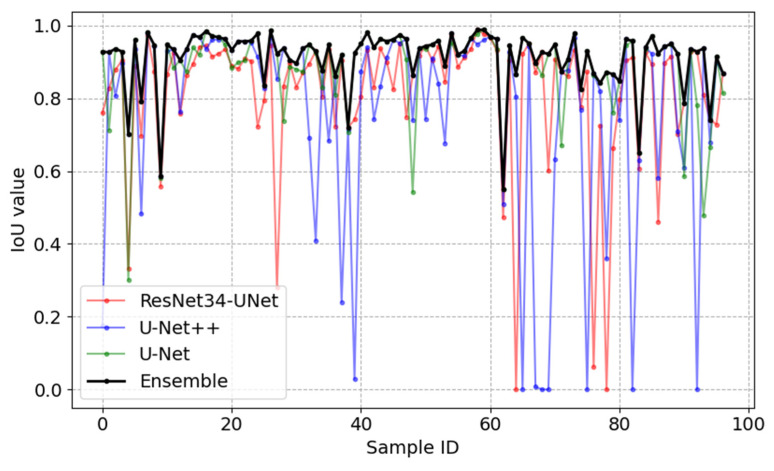
Relationship graph between the IoU value, tested image ID and segmentation model.

**Figure 12 sensors-23-03431-f012:**
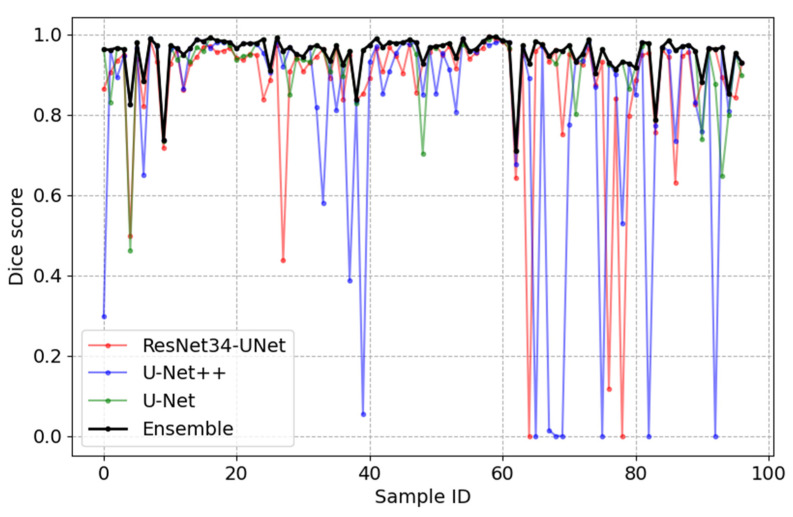
Relationship graph between the Dice score, tested image ID and segmentation model.

**Figure 13 sensors-23-03431-f013:**
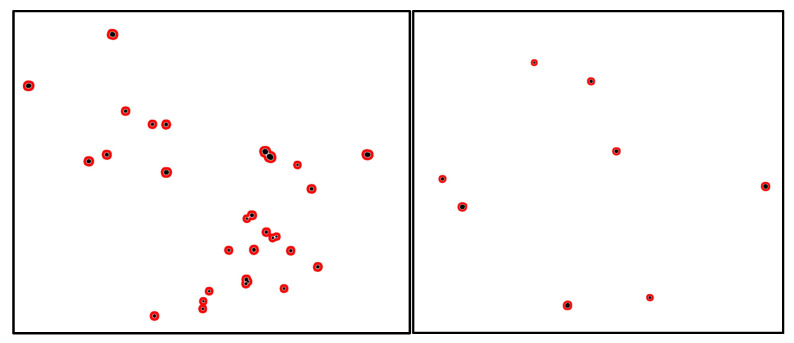
Samples of predicted whole-image segmentation maps reconstructed from patches acquired using the ensemble-based model.

**Table 1 sensors-23-03431-t001:** The evaluation results of three neural networks and the ensemble-based model.

Performance Metric	U-Net	RestNet34-UNet	U-Net++	Ensemble
IoU	0.88	0.81	0.76	**0.91**
Dice Score	0.93	0.88	0.82	**0.95**

## Data Availability

The data presented in this study are available on request from the corresponding author. The data are not publicly available due to ethical issues.

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
