# Peer review of "Automatic Detection of Microaneurysms in Fundus Images Using an Ensemble-Based Segmentation Method"

_sensors, 2023, doi:10.3390/s23073431_

Round 1

Reviewer 1 Report

The paper introduces the ever so important subject of image automatic segmentation in the biomedical field.

The paper is well presented and worked out.

I would like to see the demographics of the used dataset, which seems to be missing.

In the proposed segmentation method, it is not clear the roles and importance of the applied up-sampling and down-sampling. Please elaborate on this.

The authors should also elaborate a bit on the differences on the F1 score and Jacard index in their study.

Figure 13 is rather confusing. The left and right side should be labelled.

The rational for the ensemble-based method constituents should be mentioned.

Author Response

Thank you constructive review. The response to comments are given in attached file. 

Author Response

Response to Reviewer 2 Comments

Thank you for your constructive and perceptive review. We authors will gratefully improve our research paper based-on your recommendations. Here are our responses:

  1. The method seems a simple average among segmentation masks provided by state-of-the art network. In my opinion different solution can be evaluated. For example, a weighted average can be interesting, maybe giving a higher weight to a method with better performance.

Response 1:

Thank you for the perfect question. Weighted averaging approach were considered in the beginning research work. It is logically to think higher weights should receive segmentation model that showed better performance. However, we noticed that different segmentation models have different sensitivity to location of microaneurysm. For example, U-Net and ResNet34-Unet models are more sensitive to microaneurysms that are closer to visible blood vessels, U-Net++ is more sensitive to microaneurysms that are closer to exudates (other symptoms of DR). We decided to use average segmentation map and use a certain confidence threshold.    

  1. The ensemble operation can be included in the network topology and, therefore, in network training. In this manner is the network itself to decide the better strategy to combine mask provided by the different approaches.

Response 2:

Authors thanks for the interesting proposal. It would be incredible interesting to do performance comparison between current solution and proposed one. Unfortunately, we cannot provide this analysis in current research, because we are limited with computational resources. Proposed network topology consisting of three different segmentation models simple will not fit in available graphical card’s memory and training on CPU will take too much time. Therefore, authors are considering to include this proposed approach in the future research.

  1. There are no info about the way the ground truth images have been obtained. Authors refer to annotated images , they have been annotated by clinics?

Response 3: Text is updated based-on the answer to comment.

Ground through images (annotated images) were prepared by the group of ophthalmologists formed from 4 different countries. Each fundus image is carefully inspected and annotated by specialists who have more than 5 years of experience.

  1. Inclusion of computational burden discussion can be interesting for the readers.

Response 4: Text is updated based on the answer to comment.

The training and testing of three independent segmentation models were performed on GPU graphical card. We used Nvidia GTX1080 Ti with 11 GB of internal memory. All three models were trained on same training data with same hyper parameters. It took around 28 hours to train the smallest U-Net segmentation model that have 7.7 million of trainable parameters.  57 hours was needed to train ResNet34-Unet segmentation model that consist of 21.6 million of trainable parameters. Current model is the largest in ensemble-based approach. It took around 37 hours to train U-Net++ segmentation model that consist of 9.1 million of parameters. In total it took 122 hours (5 days) to train all ensemble-based segmentation model.  

  1. In medical image segmentation, in case of pathological profiles, often instead of providing a binary map, a probability score can be more useful. Why do you not include such representation in one example for the proposed methods?

Response 5:

Even one microaneurysm detected in fundus image signals to medical specialist about early treatment of diabetic retinopathy. In that case such fundus images with one microaneurysm could get high probability score. In the case of multiple microaneurysms we will get again one probability score per fundus image. The main aim of proposed research work is the capabilities to detect small regions in fundus images using ensemble-based segmentation method.  Therefore, authors are thinking that current representation of results is better than one probability score. Moreover, segmentation map can be used for explainability purposes, when there is a need to understand decision made by AI method.

Reviewer 3 Report

Dear Authors,     The proposed method contains fair novelty, the paper is well written and obtained results shown its efficiency.     I have only few minor requests for improving the manuscript:     Page 2, lines 72-77 - It is not common to write grants and affiliations inside research paper (except if datasets are collected from particular institutions or something similar, as you have written in 3.1. Data set subsection). For acknowledgments there is Acknowledgment section at the end of the paper. Please remove these sentences.     Page 3, lines 93-94, Sentence is grammatically incorrect, please rewrite.   Page 3, sentence in line 132 is grammatically incorrect, please rewrite.   Page 3, lines 114-115, I am not able to understand this sentence, please rewrite to make its content correct.   Page 4;5, line 168; 188: change figure 4 to Figure 4. Throughout the whole manuscript this should be fixed. Figures and Tables should always be written with capital letters.     Figure 4,6 I think numbers of each layers on these Figures are generic and not representative of what they should be in regard to the experiments in this work. This should be fixed. Also, Figure 6 and Figure 9 exceeds outside documents margins.     I thinks some figures are exceeding document margins. Please recheck.     Quality of Figure 11 and Figure 12 needs to be improved.     Page 13, Line 429- remove it.

Author Response

Thank you for constructive review. The response to comments are given in attached file. 
